# Impact of Intrinsic Muscle Weakness on Muscle–Bone Crosstalk in Osteogenesis Imperfecta

**DOI:** 10.3390/ijms22094963

**Published:** 2021-05-07

**Authors:** Victoria L. Gremminger, Charlotte L. Phillips

**Affiliations:** 1Department of Biochemistry, University of Missouri, Columbia, MO 65211, USA; vlghx9@mail.missouri.edu; 2Department of Child Health, University of Missouri, Columbia, MO 65212, USA

**Keywords:** osteogenesis imperfecta, muscle bone crosstalk, musculoskeletal disorders, myokine, osteokine, mechanotransduction

## Abstract

Bone and muscle are highly synergistic tissues that communicate extensively via mechanotransduction and biochemical signaling. Osteogenesis imperfecta (OI) is a heritable connective tissue disorder of severe bone fragility and recently recognized skeletal muscle weakness. The presence of impaired bone and muscle in OI leads to a continuous cycle of altered muscle–bone crosstalk with weak muscles further compromising bone and vice versa. Currently, there is no cure for OI and understanding the pathogenesis of the skeletal muscle weakness in relation to the bone pathogenesis of OI in light of the critical role of muscle–bone crosstalk is essential to developing and identifying novel therapeutic targets and strategies for OI. This review will highlight how impaired skeletal muscle function contributes to the pathophysiology of OI and how this phenomenon further perpetuates bone fragility.

## 1. Osteogenesis Imperfecta

Osteogenesis imperfecta (OI), also commonly known as “brittle bone disease”, is a genetically and clinically heterogeneous heritable connective tissue disorder affecting roughly 1:15,000 births [1,2]. While the most striking feature of the disease is bone fragility and skeletal dysplasia, more recent investigations have demonstrated intrinsic muscle weakness as part of the pathophysiology [3,4,5,6], and a small number of studies have highlighted the presence of metabolic perturbations in OI as well [7,8]. These studies have demonstrated evidence of a hypermetabolic state in both OI patients and mouse models including findings of elevated body temperatures and energy expenditures [7,8,9]. Other manifestations of OI include short stature, cardiopulmonary abnormalities, hearing loss, dentinogenesis imperfecta, kyphoscoliosis, and the presence of blue-gray sclera [1,2].

The heterogeneity of OI is exemplified by the more than 2000 recognized disease-causing mutations [10,11]. Roughly 85% of OI cases are the result of autosomal dominant variants resulting in qualitative (dominant negative) or quantitative (haplo-insufficient) defects in type I collagen due to mutations in the type I collagen genes, *COL1A1* and *COL1A2* [1,2,10,12]. The remaining approximate 15% of cases are the result of either autosomal dominant, autosomal recessive or X-linked mutations in genes implicated in bone mineralization, posttranslational modifications, folding, and secretion of type I collagen, as well as those involved in osteoblast maturation and function [1,2,13].

As might be expected from the vast number of disease-causing OI mutations affecting multiple genes exhibiting unique functions, the clinical severity ranges considerably; from asymptomatic to perinatal lethality [1,2,13,14]. Classically, there are four subtypes of OI (Sillence classification; I–IV) with type I being the most mild form of the disease, type II being the most severe (resulting in perinatal lethality), type III being the most severe viable form (often resulting in the non-ambulation of patients), and type IV being of moderate severity [14]. As the genetic understanding and the number of novel OI causing mutations has expanded, there has been a shift in the classification of OI with 20 types currently recognized in the Online Mendelian Inheritance in Man (OMIM) database [13,15,16]. Despite the large number of OI types described by the OMIM database and the 2019 revision of the Nosology and Classification of Genetic Skeletal Disorders highlighting five OI types based on phenotype rather than genetic origin [17], for clinical management and genetic counseling the Sillence classification remains clinically relevant.

While currently there remains no cure for OI, its genetic and clinical heterogeneity presents a challenge for treatment, preventing a generalized use of a “one size fits all” approach. This has forced physicians to be focused more on mitigating symptoms, with current treatments predominately targeted to reducing skeletal fragility with little to no consideration of the muscle weakness [2,13]. These therapeutic approaches often use bone antiresorptive and osteoanabolic pharmaceutical compounds, physical therapy, and surgical interventions, including the use of intermedullary rodding in pediatric patients [2,13,18,19,20,21].

The current standard of treatment in OI includes administration of intravenous or oral bone antiresorptive bisphosphonate therapy, which improve bone mineral density (BMD), although their efficacy in fracture reduction remains controversial and the safety and efficacy in pediatric patients remains unclear [12,13,19,22]. Additionally, long-term use of bisphosphonates may have an overall detrimental effect on bone due to decreased bone remodeling, increased accumulation of microfractures, thus compromising the bone integrity, and potential association with osteonecrosis of the jaw [12,18,22,23,24]. Bone anabolic therapies including parathyroid hormone, anti-sclerostin, and anti-TGF-β are currently being evaluated for use in OI [18,25,26,27]. It is important to note that while all these compounds may lead to increased bone formation, they do not alter the compromised integrity of bone material.

Despite the lack of consideration for muscle health in OI, skeletal muscle weakness remains an important concern with greater than 80% of mild-to-moderate OI patients experiencing muscle weakness relative to healthy individuals [6,28,29]. Furthermore, the highly synergistic nature of bone and skeletal muscle suggest that poor muscle function, observed in OI patients and mouse models, will further compromise bone properties. A better understanding of skeletal muscle weakness in OI has the potential to lead to novel therapeutic approaches that can exploit the synergy between muscle and bone.

## 2. Skeletal Muscle Weakness and Energy Metabolism in OI

Historically, muscle weakness in osteogenesis imperfecta has been attributed to hypoactivity, with patients reporting fatigue and reduced exercise capacity in addition to muscle weakness [6,30]. Studies in the homozygous osteogenesis imperfecta murine (*oim/oim*) model, which contains a nucleotide deletion in the *Col1a2* gene resulting in nonfunctional proα2(I) collagen chains and the production of homotrimeric type I collagen [α(I)_3_] rather than normal heterotrimeric type I collagen [α1(I)_2_α2(I)], were the first to demonstrate an inherent muscle pathology as part of the pathophysiology of OI [3,31,32]. Following this novel discovery, clinicians and basic scientists began to consider and investigate muscle function in OI patients and other mouse models of OI. Of these studies, one of the most significant findings was that 80% of patients with a type I OI phenotype caused by a mutation in either the *COL1A1* or *COL1A2* gene experienced muscle force deficits when muscle force was assessed via mechanography [6,28]. Through functional dynamic muscle testing (mechanography), the average peak force in OI individuals remained significantly lower than the average peak force in normal individuals even when normalized to muscle cross-sectional area [6]. Muscle function was found to correlate with OI severity, with muscle force deficits often more severe in patients with moderate and severe types IV and III OI, respectively [29]. Interestingly, Pouliot-Laforte et al. found that children with type I OI who experienced muscle force deficits were just as active as their healthy peers, although neither group of children met the daily recommended amount of activity, suggesting that reduced physical activity is not the leading cause of muscle weakness in children with OI [28]. Over the past decade, efforts to better characterize and understand muscle weakness in patients with OI have increased. While the cause of OI muscle weakness remains unknown, numerous mechanisms continue to be investigated, including: the role of altered collagen in muscle connective tissue layers and tendon, reduced mobility in patients with OI, and altered signal transduction and mechanotransduction between muscle and bone [2,15,29].

In addition to patient studies, skeletal muscle function has also been assessed in several mouse models of OI providing useful insights into the potential mechanisms of skeletal muscle weakness, as well as allowing for evaluation of novel therapeutic options targeting muscle weakness in OI [3,15,33]. These studies have demonstrated a positive correlation between OI severity and skeletal muscle weakness [3,15,33]. Skeletal muscle weakness is absent in the *+/G610C* or “Amish” mouse model of mild OI, containing a glycine to cysteine substitution in the triple helical domain of the proα2(I) collagen chain; a mutation also found in an Old Order Amish kindred affected by OI [33,34]. Whereas in contrast, models of severe OI such as the *oim/oim* and *Col1a1^Jrt/+^* exhibit reduced activity and significant skeletal muscle weakness [3,8,35,36]. In a most recent comparison of the gastrocnemius muscle transcriptomes of two mouse models of severe OI (*oim* and *Col1a1^Jrt/+^*) relative to their wildtype (WT) controls, Moffatt et al. concluded that the OI “muscle disturbances resulting from the collagen type I mutations in the mouse models could thus be viewed as a mild form of muscle dystrophy [37]”. The idea that skeletal muscle weakness may be mutation- and severity-specific in OI warrants further investigation to better understand the mechanisms of the muscle pathology.

The *Col1a1^Jrt/+^* mouse is the first combined model of both severe-type IV OI and Ehlers Danlos Syndrome (EDS) due to a *Col1a1* gene mutation leading to skipping of exon 9 and subsequent 18 amino acid deletion in the triple helical domain [35]. Classical EDS is a genetic connective tissue disorder often arising from mutations in type V collagen resulting in skin hyperelasticity, joint hypermobility, skin fragility, and blood vessel fragility, although a small subset, referred to as arthrochalasia or EDS VII, have mutations near the amino terminal end of type I collagen (often impacting exon 6) exhibiting manifestations of both OI and EDS including fragile skin, joint hypermobility, muscle hypotonia, and osteopenia [38,39]. In addition to reduced physical activity and skeletal muscle weakness, recent evidence in the *Col1a1^Jrt/+^* mouse suggests that metabolic perturbations exist, consisting of a hypermetabolic state with increased whole-body oxygen consumption and energy expenditures, potentially contributing to the pathophysiology of the disease [8]. Furthermore, a recent study in the homozygous *oim/oim* mouse demonstrated altered energy metabolism supported by increased energy expenditures in the *oim/oim* mice [9] and by skeletal muscle mitochondrial dysfunction, as evidenced by drastically reduced gastrocnemius mitochondrial respiration rates [32]. While there are limited studies evaluating energy metabolism in OI patients, a study by Cropp and Myers in 1972 described the presence of a hypermetabolic state in adolescent male OI patients [7]. Taken together, the report of a metabolic perturbations in both the mentioned patient cohort and in the *Col1a1^Jrt/+^* and *oim/oim* models suggests that energy metabolism is an important contributor to the pathophysiology of OI. The idea of altered energy metabolism in OI may not be that surprising due to the wide prevalence of skeletal muscle weakness in patients with OI and that skeletal muscles account for roughly 20% of the basal metabolic rate [18,29,40,41]. Whether mitochondrial dysfunction is also prevalent in human OI patients and other OI mouse models warrants further investigation.

While the bone phenotype in OI has been well characterized and thoroughly evaluated in both patients and several animal models, skeletal muscle weakness in OI is a relatively recent finding and understanding the mechanisms leading to this phenomenon are just beginning to be revealed. Understanding these mechanisms is critical to potentially identifying novel therapeutic targets that will improve the observed muscle phenotype, as well as ultimately benefit bone strength through the highly synergistic muscle–bone cross-talk [42,43,44,45,46].

Cases of muscle weakness and altered energy metabolism, like those observed in OI patients and mouse models exhibiting type I collagen mutations, are also found in animal models and patients with muscular dystrophies possessing gene defects in extracellular matrix proteins such as laminin α2 (*LAMA2*) and type VI collagen (*COLVI*) [47,48]. The clinical presentation of LAMA2-associated muscular dystrophies closely resembles that of the COLVI-related Bethlem myopathy, including muscle weakness, muscle hypotonia, and joint contractures [49]. Interestingly, fibroblasts cultured from one patient with a LAMA2-related muscular dystrophy exhibited reduced COLVI secretion, although no mutations were found in their *COL6A1-3* genes, suggesting the important role of extracellular matrix homeostasis in skeletal muscle health [49]. In addition to the clinical similarities seen among these muscular dystrophies, both the *Col6a1^−/−^* and *oim/oim* mouse also exhibit mitochondrial dysfunction [9,32,50].

While there is no clear link between type I collagen and mitochondrial function, there is limited evidence of interactions between type I and type VI collagen, which may imply an indirect relationship between type I collagen and mitochondrial function. An early report demonstrates the binding of type I collagen to chicken α3(VI) collagen [51]. Furthermore, an in silico analysis of the mouse α3(VI) collagen chain (*Col6a3*) on the STRING: functional protein association network database reveals that the type I collagen genes, *Col1a1* and *Col1a2*, are amongst the top five predicted interaction partners (https://string-db.org, accessed 22 April 2021). Type VI collagen is an important extracellular matrix protein found in many diverse tissues, including skeletal muscle [52]. In skeletal muscle, the extracellular matrix plays an important role in maintaining structure and lateral force transmission throughout the muscle [53]. The collagen network of the perimysium is composed of mainly type I collagen with types III, VI, and XII collagens present in lesser amounts [54]. Defects in type I collagen may lead to destabilization of the perimysial collagen bundles and ultimately the perimysial junctional plate, a region important for lateral force transmission where the perimysial collagen bundles interact with the endomysium [54]. Moreover a recent study of teriparatide (a bone anabolic) treatment in OI patients and post-menopausal women revealed increased presence of collagen biomarkers including types II, III, IV, V, and VI in OI patients, but not in post-menopausal women [55]. This further suggests that dysregulated extracellular matrix interactions contribute to the muscle pathology in OI.

## 3. Muscle–Bone Crosstalk

Osteogenesis imperfecta translates to imperfect bone formation and, as the name literally suggests, is a bone disease. Yet, the significant number of OI patients that experience muscle force deficits suggests the wide prevalence of muscle weakness in addition to the bone fragility [6,28,29]. Thus, it is important to consider the impact that muscle weakness may have on further compromising the bones of OI patients. Until recently, the relationship between these two organs has been limited to the context of physical forces and biomechanics [45,46,56]. In this instance, the muscle acts as a pulley, exerting force on the bone, acting as a lever, to move the skeleton [56]. The understanding of this relationship has evolved over the past decade, with the realization that both bones and muscles act as secretory organs communicating via biochemical signaling in addition to mechanotransduction [45,46,56].

### 3.1. Biomechanical

The earliest understanding of the relationship between bone and muscle was that of their physical interactions, with positive correlations between muscle weight and bone mass [57]. This concept began as Wolf’s law, and eventually led to Frost’s mechanostat theory, which postulates that bone strength is tightly coupled to skeletal muscle action [42,58,59]. Significant evidence in the field of musculoskeletal health has corroborated this initial conceptualization from 1892 [59,60]. Countless studies evaluating the consequences of disuse or unloading on these two organs support the notion that muscle force and bone strength are tightly coupled [5,15,56,61,62,63,64,65].

The ability of bone to remodel its geometry and size in response to the mechanical strain from muscle is primarily facilitated by the osteocyte, a mechanosensory cell [66]. Osteocytes arise from osteoblasts (bone-forming cells), comprise up to 95% of all bone cells, and are known for their roles in orchestrating bone remodeling via control of osteoblasts and osteoclasts (bone-resorbing cells), as well as in mechano-sensation [66]. Osteocytes are embedded in the bone matrix and have an average half-life of 25 years. Osteocytes possess dendritic processes which aid in communication with other osteocytes, osteoblasts, and osteoclasts [66,67]. For mechanotransduction to occur, the osteocyte must perceive a mechanical signal and then convert it to a biochemical signal to elicit a response by osteoblasts and/or osteoclasts to facilitate bone remodeling [66,68,69,70]. There is evidence that the dendritic processes, cell body, and cilium contribute to the osteocyte’s ability to function as a mechanosensory cell [66,69]. However, the exact mechanisms by which the osteocytes act as mechanosensing cells remains poorly understood. It is hypothesized that the osteocyte can respond to several factors in response to mechanical strain, including the physical deformation of the bone matrix, fluid flow shear stress, and streaming potentials [66,69,71].

In response to a mechanical stimulus, intracellular calcium levels become increased and the osteocyte begins to regulate and orchestrate osteoblast and osteoclast activity to control bone remodeling [69]. Mechanical stress induces an increase in intracellular calcium, leading to the opening of voltage-operated calcium channels allowing the osteocyte to release signals including nitrous oxide, prostaglandins, and ATP, which are believed to have direct effects on osteoblasts and osteoclasts [69]. Prostaglandins, nitrous oxide, and ATP have a direct positive effect on bone formation, while nitrous oxide has a negative effect on osteoclasts, resulting in a net gain of bone [69]. Activation of the Wnt/β-catenin pathway is one of the most important signaling pathways modulated by the osteocyte’s response to mechanical stress, resulting in increased bone formation [72,73]. Furthermore, members of the Wnt/β-catenin signaling pathway, including LRP5/6 and Wnt ligand, are inhibited via sclerostin, dickkopf-1 (DKK1), and selected frizzled-related protein 1 (sFRP1) [69,74]. Receptor activator of nuclear kappa-B ligand (RANKL), another important regulator in bone remodeling, activates the osteoclast resulting in increased bone resorption, whereas osteoprotegerin (OPG) inhibits RANKL, resulting in increased bone formation [69,74]. Finally, macrophage colony stimulating factor (M-CSF) regulates differentiation of osteoclasts from hematopoietic stem cells [74]. Importantly, several of these signaling molecules serve as targets of novel therapies being investigated in OI, including the use of anti-sclerostin and RANKL inhibitor antibodies [18].

As previously mentioned, biomechanical interactions are key contributors to muscle–bone crosstalk and health, and thus implicate the importance of physical activity on bone health. Fifty percent of peak adult bone mass is accrued during adolescence and there is strong evidence that 20–40% of adult peak bone mass is determined by modifiable factors including exercise, further reiterating the importance of physical activity in children with OI [75,76]. Studies have demonstrated that children with OI have activity levels comparable to those of their healthy peers, although exercise tolerance is reduced in patients with type I OI [28,30,70]. There is evidence that increased physical activity may also improve aerobic capacity and muscle strength in patients with OI [28,30,70,77]. The use of exercise and whole body vibration therapy in OI has been investigated as modalities of treatment to exploit the biomechanical relationship between muscle and bone [15].

Whole-body vibration therapy (WBV) has been implemented as a treatment strategy in OI and is an especially attractive target for children with severe OI and lower ambulation and ability to participate in traditional physical activities [78,79,80]. Studies evaluating WBV therapy using the *oim* mouse demonstrated improvement of several bone parameters, including increased cortical bone area, cortical thickness, and trabecular bone volume, while no significant improvement in bone mechanical properties were observed [79]. Limited WBV studies in OI patients exist and their results have been mixed: Hoyer-Kuhn et al. demonstrated improved bone mineral density and motor function in patients with OI, while Högler et al. found increased lean mass, but no changes in bone mass or muscle function [78,81].

A recent study by Berman et al. evaluated the osteogenic potential and tensile strain in vivo on *oim/oim* mouse tibias during peak tetanic muscle contraction and demonstrated that the tensile strain on the *oim/oim* tibia was 41% less than that of WT mouse tibias, although it was still within a range to elicit osteogenesis [82]. Additionally, this study reported the high prevalence of Achilles tendon calcification and calcaneus fractures in the *oim/oim* animals, which also likely further impairs muscle–bone crosstalk [82]. These findings cannot differentiate whether the *oim/oim* bone is less responsive to mechanical forces elicited by muscle or whether this reflects the altered mechanical properties of tendons likely impairing the transmission of force from skeletal muscle to bone [3,83]. Regardless, these data suggest that *oim/oim* bone may require a higher frequency or duration of stimulation to elicit higher levels of osteogenesis in response to muscle contraction.

The discrepancy and inability of some OI patients to respond to WBV may be the result of genetic mutation specificity, differing levels of OI severity, and/or may reflect a reduced biomechanical responsiveness of OI bone [81]. Regardless, the use of exercise and WBV as therapeutic approaches in OI are attractive targets that are minimally invasive and cost-effective ways to potentially exploit the biomechanical relationship of bone and muscle.

### 3.2. Biochemical

The nature of muscle–bone crosstalk in the past was thought to be limited to the biomechanical relationship. However, recent investigations have highlighted the presence of extensive biochemical crosstalk that occurs with both bone and muscle acting as secretory and endocrine organs [45,46,84]. One of the earliest reports suggesting biochemical crosstalk between muscle and bone, specifically a positive effect of muscle tissue on osteogenesis (independent of contraction), demonstrated improved fracture healing (increased speed and quality) when an open tibial fracture was wrapped with a muscle flap relative to non-wrapped tibias in a canine model [85]. Additionally, similar studies in rodent models have reproduced the improved tibia open fracture healing in the presence of a muscle flap as compared to a fasciocutaneous flap [86]. Since these findings, the field of musculoskeletal interaction research has grown exponentially with roughly 600 articles published in 2019 [87].

Proposed musculoskeletal biochemical interactions can be summarized into three main mechanisms [87]. The first two mechanisms suggest that, due to their highly vascularized natures, both bone and muscle secrete soluble factors into the blood stream [87]. The third mechanism consists of direct cell-to-cell contact; due to the close proximity of bone and muscle, it is likely that small molecules are able to be directly signaled between the two organs via extracellular vesicles or passive diffusion [87]. These factors secreted by muscle and bone are referred to as myokines and osteokines, respectively.

Myostatin, discovered in 1997, was the first and remains the best characterized myokine, although the term myokine was not coined until 2003, by Pedersen et al. [84,87,88,89]. Since then, over 3000 myokines have been identified [90]. Additionally, there is evidence that different myokines may be preferentially secreted by certain types of muscle fibers (i.e., glycolytic vs. oxidative) [90,91]. Some of the most prominent myokines, especially in the context of muscle–bone crosstalk, include: myostatin (mstn), interleukin-6 (IL-6), irisin, and β-aminoisobutyric acid (BAIBA) [15,46,87,91]. Just as muscle acts as a secretory organ releasing myokines that will act on bone, bone reciprocates this action, releasing osteokines that act on muscle including: osteocalcin, Wnt3a, TGF-β, and sclerostin [15,87]. The role of these myokines and osteokines are summarized in Table 1 and Table 2, respectively. Several examples of altered biochemical signaling pathways in OI muscle and bone have been described and novel therapeutics have emerged acting on these altered pathways [2,13,18,20].

The role of these osteokines in osteogenesis imperfecta has been actively pursued, while fewer studies exist regarding the role of different myokines in the pathology of OI. Of the myokines, myostatin has been the most studied in the context of OI. Myostatin is a negative regulator of muscle mass that signals through the activin receptor type IIB to inhibit muscle growth [111]. Several studies in the *oim*, *+/G610C*, and *Col1a1^Jrt/+^* mouse models have highlighted the potential of myostatin inhibition having a beneficial outcome on bone properties [31,112,113,114]. The first report highlighting the benefit of myostatin (mstn) inhibition demonstrated that *+/oim* mice, modeling a mild-to-moderate patient OI, when also deficient in mstn (*+/oim +/mstn*), exhibited improved femoral bone biomechanical properties compared to their *+/oim* littermates [112]. Studies using activin receptor IIB decoy molecules suggest that the responsiveness to myostatin inhibition in OI may be mutation-specific. Specifically, in a large study with both +/*G610C* and *oim/oim* mice on the same congenic C57BL/6 J background, activin receptor IIB decoy molecule (RAP-031; Acceleron Pharma, Inc., Cambridge, MA, USA) treatment resulted in improved skeletal, but not muscle, properties in the *+/G610C* mouse, modeling a mild-to-moderate human OI, and improved skeletal muscle contractile, but not skeletal properties in the *oim/oim* mouse [31,113]. The activin receptor IIB decoy molecule (RAP-031) used in the *+/G610C* and *oim/oim* studies neutralizes activin and myostatin, but is also thought to neutralize BMP9/10. The ability to bind BMP9/10 was implicated in the vascular side effects seen in human trials using the human activin receptor IIB decoy molecule equivalent (ACE-031; Acceleron Pharma, Inc., Cambridge, MA, USA) [115,116]. In a separate study, utilizing a different activin receptor IIB decoy molecule (ACE-2494; Acceleron Pharma, Inc., Cambridge, MA, USA) which does not neutralize BMP9/10, in the *Col1a1^Jrt/+^* mouse led to increased muscle mass, bone mass, and improved bone geometry [114]. Additionally, a recent study treating *+/G610C* and WT mice with a monoclonal anti-myostatin antibody (anti-mstn) demonstrated increased muscle weights in all the mice treated with anti-mstn relative to those treated with control antibody, regardless of genotype or sex, while bone parameters improved only in WT male mice [117]. Furthermore, there is evidence that maternal myostatin inhibition may have a lasting, positive impact on offspring musculoskeletal health via developmental programming, as *+/oim* pups born to *+/mstn*-deficient dams had improved bone biomechanical properties at 4 months of age [118]. These studies suggest that myostatin inhibition may serve as a novel therapeutic target in the treatment of OI, although responses may be drug-, dose-, and mutation-specific.

As indicated above, relative to the myokines, there have been more investigations with regard to osteokines in OI. Osteocalcin levels are elevated in both patients and the *Col1a1^+/Jrt^* OI mouse model [8,119,120]. Additionally, dysregulated TGF-β signaling has been observed in OI mouse models with increased signaling observed in both recessive *(Crtap^−/−^)* and autosomal dominant (*+/G610C* and *Col1a1^+/Jrt^*) variants of the disease [18,25,121]. However, treatment with a TGF-β neutralizing antibody in the two separate studies only improved bone properties in two of these models (*Crtap^−/−^* and *Col1a1^+/Jrt^*), suggesting the importance of mutation-specific treatment approaches [25,121]. Clinical trials in OI patients are also ongoing using TGF-β antibody [2]. Finally, there is no clear consensus on whether or not the levels of sclerostin are altered in OI patients, although there is evidence of improved bone properties in several mouse models of OI (*oim/oim* [27], *+/G610C*, *Brtl^−/−^* [122], *Col1a1^+/Jrt^* [123], and *Crtap^−/−^* [124]) subjected to sclerostin inhibition. Patient clinical trials utilizing anti-sclerostin antibodies are currently ongoing with initial results showing increased markers of bone formation and reduced markers of bone resorption [2,15,18,125,126].

The relationship between bone and muscle is highly synergistic, exhibiting both biomechanical and biochemical crosstalk. Biomechanical and biochemical crosstalk are not mutually exclusive. The role of myostatin in muscle–bone crosstalk highlights this point. Inhibition of the myokine, myostatin leads to increased muscle size and force, which is hypothesized to act through mechanotransduction to increase bone mass. Myostatin inhibition is also hypothesized to act directly through biochemical signaling, to downregulate osteoclastogenesis [94,95]. Understanding the complexity of muscle–bone crosstalk in OI is crucial to determining novel therapies that can exploit this relationship by improving muscle function to ultimately improve bone strength.

## 4. Conclusions

Muscle–bone crosstalk is an important contributor to overall musculoskeletal health, and impaired muscle function in OI patients may further compromise their already compromised skeletal integrity. Muscle and bone communicate with one another via both mechanotransduction and biochemical signaling. Improving muscle strength and exploiting the muscle–bone crosstalk through exercise/WBV and novel drug therapies, respectively, promises to lead to improved patient outcomes and quality of life in OI. Although the studies regarding exercise in OI are limited, they suggest that children with type I OI tend to be just as active as their healthy peers, even though they exhibit reduced muscle strength [28]. This, along with the recent study suggesting that mice with OI require greater forces on bone to elicit an osteogenic response [82], further emphasize the need to explore different targeted physiotherapeutic modalities for the potential treatment of OI. To better understand the complex relationship between bone and muscle in OI, mechanistic studies evaluating skeletal muscle weakness and its impact on bone health as well as the impact of poor bone health on skeletal muscle function are needed. Energy metabolism has recently become an area of interest among the OI research community and should continue to be thoroughly evaluated to determine its role in OI bone and muscle weakness.

## Figures and Tables

**Table 1 ijms-22-04963-t001:** Myokines and their roles on bone.

Myokine	Role in Bone
Myostatin (mstn)	A member of the TGF-β superfamily and negative regulator of muscle mass. In addition to its inhibitory effect on muscle growth, mstn has been repeatedly shown to have a negative impact on bone formation via increased osteoclastogenesis, increased osteocyte expression of negative regulators of bone including sclerostin, and reduced expression of osteoblast differentiation markers [84,87,89,92,93,94].
Interleukin-6 (IL-6)	Although secreted by multiple tissues, large amounts of IL-6 are secreted by muscle in response to exercise leading to its characterization as a myokine [88]. Interestingly, IL-6 has been shown to upregulate both osteoclast and osteoblast formation [87,95,96].
Irisin	Irisin has been shown to be positively correlated with BMD and negatively correlated with serum sclerostin [97,98]. Additionally, in mice, treatment with recombinant irisin led to improved bone geometry and mechanical properties [99].
β-aminoisobutyric acid (BAIBA)	BAIBA, a myokine secreted in response to exercise, was originally identified for its function in the browning of white adipose tissue, though more recent studies have demonstrated a role in osteocytes as a protective agent against reactive oxygen species [100,101].

**Table 2 ijms-22-04963-t002:** Osteokines and their roles in muscle.

Osteokine	Role in Muscle
Osteocalcin (OCN)	OCN, secreted by the osteoblast, plays an important role in glucose and energy homeostasis and is believed to promote nutrient catabolism and uptake in skeletal muscle as well as improve exercise capacity [102,103,104].
Wnt3a	Wnt3a, secreted by osteocytes, was shown to promote myogenesis via increased myogenin and myoD expression in a cultured osteocyte cell line (C2C12) [105,106].
Transforming growth factor-β (TGF-β)	TGF-β, an osteokine secreted by osteoblasts, has been shown to negatively impact skeletal muscle function via calcium leakage and increased oxidative stress [107,108].
Sclerostin	Although there is not evidence of sclerostin, secreted by the osteocyte, directly effecting muscle, it has been shown to inhibit Wnt-3a action in skeletal muscle, thus indirectly negatively impacting muscle [105,109,110].

## Data Availability

Data sharing not applicable. No new data were created or analyzed in this review.

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
