# Peer review of "Impact of Intrinsic Muscle Weakness on Muscle–Bone Crosstalk in Osteogenesis Imperfecta"

_ijms, 2021, doi:10.3390/ijms22094963_

Round 1

Reviewer 1 Report

This is a review article on the muscle phenotype of osteogenesis imperfecta (OI) and the muscle-bone relationship in this disorder. The review is very well written and provides an up-to-date overview of the relevant literature. A few minor issues could be clarified.

Specific Comments

  1. Line 85: The statement that ‘greater than 80% of patients with mild to moderate OI experience muscle force deficits’ does not seem an accurate summary of the cited articles. These articles rather seem to suggest that 80% of OI patients had a numerically lower force test result than their respective age- and sex-matched healthy peer. Having a slightly lower muscle force than a matched healthy control does not necessarily mean that you have a deficit.
  2. Section ‘Skeletal muscle weakness and energy metabolism in OI’: The section nicely summarizes the evidence that some muscle function tests yield lower average results in OI patients than in healthy control groups. It would be interesting to put such findings into some clinical context. How severe is the muscle weakness of OI when compared to ‘established’ muscle disorders caused by mutations in genes that code for extracellular matrix components such as Bethlem myopathy or LAMA2 related muscle dystrophies? What are the clinical symptoms related to muscle weakness that OI patients experience?
  3. In Figure 1, the word ‘mechanotransduction’ is written next to the muscle in a way that suggests that muscle is responsible for mechanotransduction. However, mechanotransduction is a process that occurs in the bones as a response to the forces produced by muscles. It would be more appropriate to write ‘Force’ next to the muscle and ‘mechanotransduction’ at the site where the muscle attaches to the bone.
  4. Line 119: Was this an ‘old Amish’ kindred or an ‘Old Order Amish’ kindred?

Author Response

  1. Reviewer #1:

  1. Comment: “Line 85: The statement that ‘greater than 80% of patients with mild to moderate OI experience muscle force deficits’ does not seem an accurate summary of the cited articles. These articles rather seem to suggest that 80% of OI patients had a numerically lower force test result than their respective age- and sex-matched healthy peer. Having a slightly lower muscle force than a matched healthy control does not necessarily mean that you have a deficit.”

Response: We appreciate the reviewer’s comment regarding concern over the use of the term deficit. Please see that we have changed the statement to now read:

Lines 92-94 : “Despite the lack of consideration for muscle health in OI, skeletal muscle weakness remains an important concern with greater than 80% of mild to moderate OI patients experiencing muscle weakness relative to healthy individuals (Veilleux, Lemay, et al.; Pouliot-Laforte et al.; Veilleux, Trejo, et al.)”

  1. Comment: Section ‘Skeletal muscle weakness and energy metabolism in OI’: The section nicely summarizes the evidence that some muscle function tests yield lower average results in OI patients than in healthy control groups. It would be interesting to put such findings into some clinical context. How severe is the muscle weakness of OI when compared to ‘established’ muscle disorders caused by mutations in genes that code for extracellular matrix components such as Bethlem myopathy or LAMA2 related muscle dystrophies? What are the clinical symptoms related to muscle weakness that OI patients experience?

Response: We appreciate the reviewer’s thoughtful comment and have elaborated on this section to include more information comparing muscular dystrophies and OI highlighting the potential role of the extracellular matrix. Please see our revisions below:

Lines 133-136: “In a most recent comparison of the gastrocnemius muscle transcriptomes of two mouse models of severe OI (oim and Col1a1Jrt/+) relative to their wildtype (WT) controls Moffatt et al concluded that the OI “muscle disturbances resulting from the collagen type I mutations in the mouse models could thus be viewed as a mild form of muscle dystrophy(Moffatt et al.).”

Lines 168-196:  “Cases of muscle weakness and altered energy metabolism, like those observed in OI patients and mouse models exhibiting type I collagen mutations, are also found in animal models and patients with muscular dystrophies possessing gene defects in extracellular matrix proteins such as laminin α2 (LAMA2) and type VI collagen (COLVI) (Sarkozy et al.; Bönnemann). The clinical presentation of LAMA2 associated muscular dystrophies closely resembles that of the COLVI related Bethlem myopathy including muscle weakness, muscle hypotonia, and joint contractures(Nelson et al.). Interestingly, fibroblasts cultured from one patient with a LAMA2 related muscular dystrophy exhibited reduced COLVI secretion, although no mutations were found in their COLVIA1-3 genes, suggesting the important role of extracellular matrix homeostasis in skeletal muscle health (Nelson et al.). In addition to the clinical similarities seen among these muscular dystrophies, both the COLVI-/- and oim/oim mouse also exhibit mitochondrial dysfunction(Irwin et al.; V.L. Gremminger et al.; Victoria L. Gremminger et al.).

While there is no clear link between type I collagen and mitochondrial function, there is limited evidence of interactions between type I and type VI collagen, which may imply an indirect relationship between type I collagen and mitochondrial function. An early report demonstrates the binding of type I collagen to chicken α3(VI) collagen (Bonaldo et al.). Furthermore, an in silico analysis of the mouse α3(VI) collagen chain (Col6a3) on the STRING: functional protein association network database reveals that the type I collagen genes, Col1a1 and Col1a2, are amongst the top five predicted interaction partners (https://string-db.org). Type VI collagen is an important extracellular matrix protein found in many diverse tissues including skeletal muscle (Lamandé and Bateman). In skeletal muscle, the extracellular matrix plays important role in maintaining structure and lateral force transmission throughout the muscle (Csapo et al.). The collagen network of the perimysium is composed of mainly type I collagen with types III, VI, and XII collagens present in lesser amounts (Passerieux et al.). Defects in type I collagen may lead to destabilization of the perimysial collagen bundles and ultimately the perimysial junction plate, a region important for lateral force transmission where the perimysial collagen bundles interact with the endomysium (Passerieux et al.). Moreover a recent study of teriparatide (a bone anabolic) treatment in OI patients and post-menopausal women revealed increased presence of collagen biomarkers including types II, III, IV, V, and VI in OI patients, but not in post-menopausal women (Nicol et al.). This further suggests dysregulated extracellular matrix interactions contribute to the muscle pathology in OI.” 

  1. Comment: In Figure 1, the word ‘mechanotransduction’ is written next to the muscle in a way that suggests that muscle is responsible for mechanotransduction. However, mechanotransduction is a process that occurs in the bones as a response to the forces produced by muscles. It would be more appropriate to write ‘Force’ next to the muscle and ‘mechanotransduction’ at the site where the muscle attaches to the bone.

Response: Thank you for this suggestion and, as per the recommendation of Reviewer 2, we have corrected and redesigned our Figure 1.

  1. Comment: Line 119: Was this an ‘old Amish’ kindred or an ‘Old Order Amish’ kindred?

Response: We appreciate the reviewer pointing out this mistype and have corrected to read Old Order Amish kindred (Lines 130-131).

Reviewer 2 Report

Muscle health in OI is still a relatively unexplored subject that deserves scientific attention. The review addresses this subject by elaborating on the bone-muscle crosstalk in the context of OI. The authors are suggested to clarify/adjust the following:

Figure 1: please improve quality.

Introduction

  • “The remaining approximate 15% of cases are the result…”: in this paragraph it is good to make a distinction between autosomal dominant and recessive genes.
  • “…there has been a shift in the classification of OI with 20 types currently recognized in the Online Mendelian…”: maybe here it can be additionally mentioned that the Sillence classification (with a note of the mutation) is the most practical/recommended to communicate OI, considering that the 20 OI types show clinical overlap and present a far more complicated system.
  • “…have highlighted the presence of a metabolic phenotype in OI as well…”: please explain shortly the metabolic phenotype.  

Skeletal muscle weakness and energy metabolism in OI

  • “…muscle weakness, recent evidence in the Col1a1Jrt/+ mouse suggests a metabolic phenotype…”: please explain what is meant by metabolic phenotype.
  • “…exhibiting manifestations of both OI and EDS including fragile skin, joint hypermobility, and osteopenia…”: and muscle hypotonia.

Biomechanical

  • “…osteocyte begins to regulate and integrate osteoblasts and osteoclasts to control bone…”: it is not clear what is meant by integrate.
  • “With one of the most important signaling pathways modulated by the osteocyte’s response to mechanical stress being the Wnt/β-catenin pathway”: please adjust sentence structure.
  • “Furthermore, members of the Wnt/β-catenin signaling pathway, and thus bone formation, are inhibited via sclerostin, dickkopf-1 (DKK1), and selected frizzled-related protein 1 (sFRP1)”: please clarify sentence.
  • “In response to a mechanical stimulus, intracellular calcium levels become increased and the osteocyte begins to regulate and integrate osteoblasts and osteoclasts to control bone remodeling”: in this paragraph it would be enlightening to also mention the cell  source of these signaling molecules.

 Biochemical

  • “Other data suggest that the responsiveness to myostatin inhibition is OI mutation specific with studies using the activin receptor IIB decoy molecule resulting in improved skeletal properties in the +/G610C mouse, modeling a mild to moderate human OI, and improved skeletal muscle contractile properties in the oim/oim mouse”: it is not completely clear why it is mutation specific. Were both measured in both models?
  • “Whereas the activin receptor IIB decoy molecule used in the +/G610C and oim/oim studies neutralizes BMP9/10 in addition to activin and myostatin, a different activin receptor IIB decoy molecule that does not inhibit BMP9/10 was used in the Col1a1Jrt/+ mouse”: it is not clear what is the significance of this comparison.
  • “These studies suggest that myostatin may serve as a novel therapeutic target in the treatment of OI although responses may be dose and mutation specific.”: it is perhaps difficult to say if it is mutation specific or not if the different mouse models have not been examined in a uniform manner.
  • “The role of myostatin in muscle-bone crosstalk highlights this point. Inhibition of the myokine, myostatin, leads to altered mechanotransduction via increased muscle size in addition this inhibition is hypothesized to have a direct effect on biochemical signaling, as well, leading to a downregulation of osteoclast”: please improve sentence structure and content.

Conclusions

  • Is there a recommendation that can be made about physical exercise in OI based on the review findings?

General

  • It would add to also discuss the production of collagen from osteoblasts as a result of physical activity which can affect bone quality.
  • The paragraph “The role of these osteokines in osteogenesis imperfecta has been actively….”until “...that can exploit this relationship by improving muscle function to ultimately improve bone strength.” seems to have been written in a less structured, clear and concise manner compared to the rest of the article. In several places it is not clear what the significance of the stated literature is. Please revise accordingly.
  • There is extensive description of the bone on the cellular level but this is absent for the muscles.
  • The title is “Impact of intrinsic muscle weakness on muscle-bone crosstalk in osteogenesis imperfecta”: please elaborate more on the conclusions that can be made about the intrinsic properties of OI muscle; the article seems to focus a lot on the muscle-bone interaction, which is important in OI pathophysiology but not completely in line with the title. Also please include what is the role of collagen in muscle structure and function.

Author Response

  1. Reviewer #2

  1. Comment: Figure 1: please improve quality.

Response: Please see we have taken into consideration both Reviewer’s comments and have revised Figure 1.

  1. Comment: Introduction
  1. “The remaining approximate 15% of cases are the result…”: in this paragraph it is good to make a distinction between autosomal dominant and recessive genes.
  2. “…there has been a shift in the classification of OI with 20 types currently recognized in the Online Mendelian…”: maybe here it can be additionally mentioned that the Sillence classification (with a note of the mutation) is the most practical/recommended to communicate OI, considering that the 20 OI types show clinical overlap and present a far more complicated system.
  3. “…have highlighted the presence of a metabolic phenotype in OI as well…”: please explain shortly the metabolic phenotype.

Response: We thank the reviewer for these comments, which we believe strengthen the introduction of our manuscript. Please see that we have addressed each point below:

  1. Lines 55-61: “Roughly 85% of OI cases are the result of autosomal dominant variants resulting in qualitative (dominant negative) or quantitative (haplo-insufficient) defects in type I collagen due to mutations in the type I collagen genes, COL1A1 and COL1A2(Marom et al.; Marini et al.; Dalgleish; Ralston and Gaston). The remaining approximate 15% of cases are the result of either autosomal dominant, autosomal recessive or X-linked mutations in genes implicated in bone mineralization, posttranslational modifications, folding, and secretion of type I collagen, as well as those involved in osteoblast maturation and function (Etich et al.; Marom et al.; Marini et al.).”
  2. Lines 70-74: “Despite the large number of OI types described by the OMIM database and the 2019 revision of the Nosology and Classification of Genetic Skeletal Disorders highlighting five OI types based on phenotype rather than genetic origin (Mortier et al.), for clinical management and genetic counseling the Sillence classification remains clinically relevant.”
  3. Lines 48-51: “…highlighted the presence of metabolic perturbations in OI as well (Cropp and Myers; Boraschi-Diaz et al.). These studies have demonstrated evidence of a hypermetabolic state in both OI patients and mouse models including findings of elevated body temperatures and energy expenditures (Cropp and Myers; Boraschi-Diaz et al.; Victoria L. Gremminger et al.).”

  1. Comment: Skeletal muscle weakness and energy metabolism in OI
  1. “…muscle weakness, recent evidence in the Col1a1Jrt/+ mouse suggests a metabolic phenotype…”: please explain what is meant by metabolic phenotype.
  2. “…exhibiting manifestations of both OI and EDS including fragile skin, joint hypermobility, and osteopenia…”: and muscle hypotonia.

Response: We are grateful to the reviewer for these recommendations and have elaborated on these points as follows:

  1. Lines 146-150“joint hypermobility, muscle hypotonia, and osteopenia (Bowen et al.; Miklovic and Sieg). In addition to reduced physical activity and skeletal muscle weakness, recent evidence in the Col1a1Jrt/+ mouse suggests metabolic perturbations exist, consisting of a hypermetabolic state with increased whole body oxygen consumption and energy expenditures, potentially contributing to the pathophysiology of the disease (Boraschi-Diaz et al.).
  2. Line 146: We have added muscle hypotonia to this description as well.

  1. Comment: Biomechanical
  1. “…osteocyte begins to regulate and integrate osteoblasts and osteoclasts to control bone…”: it is not clear what is meant by integrate.
  2. “With one of the most important signaling pathways modulated by the osteocyte’s response to mechanical stress being the Wnt/β-catenin pathway”: please adjust sentence structure.
  3. “Furthermore, members of the Wnt/β-catenin signaling pathway, and thus bone formation, are inhibited via sclerostin, dickkopf-1 (DKK1), and selected frizzled-related protein 1 (sFRP1)”: please clarify sentence.
  4. “In response to a mechanical stimulus, intracellular calcium levels become increased and the osteocyte begins to regulate and integrate osteoblasts and osteoclasts to control bone remodeling”: in this paragraph, it would be enlightening to also mention the cell source of these signaling molecules.

Response: We appreciate the reviewer’s thoughtful comments regarding the clarity of these statements. Please see that we have deliberately considered each point and have made the following clarifications:

  1. Lines 231-232: Due to the unclear meaning of integrate, we have modified the sentence to read “…response to a mechanical stimulus, intracellular calcium levels become increased and the osteocyte begins to regulate and orchestrate osteoblast and osteoclast activity to control bone remodeling (Dallas et al.).”
  2. Lines 238-240: We have improved the sentence structure now reading, “Activation of the Wnt/β-catenin pathway is one of the most important signaling pathways modulated by the osteocyte’s response to mechanical stress, resulting in increased bone formation (Tu et al.; Glass and Karsenty).”
  3. Lines 240-242: “Furthermore, members of the Wnt/β-catenin signaling pathway including LRP5/6 and Wnt ligand, are inhibited via sclerostin, dickkopf-1 (DKK1), and selected frizzled-related protein 1 (sFRP1)(Duan and Bonewald; Dallas et al.).”
  4. Lines 233-236: Please see that we have elaborated on this topic, “Mechanical stress induces an increase in intracellular calcium leading to the opening of voltage operated calcium channels allowing the osteocyte to release signals including nitrous oxide, prostaglandins, and ATP ,which are believed to have direct effects on osteoblasts and osteoclasts (Dallas et al.).”

  1. Comment: Biochemical
  1. “Other data suggest that the responsiveness to myostatin inhibition is OI mutation specific with studies using the activin receptor IIB decoy molecule resulting in improved skeletal properties in the +/G610C mouse, modeling a mild to moderate human OI, and improved skeletal muscle contractile properties in the oim/oim mouse”: it is not completely clear why it is mutation specific. Were both measured in both models?
  2. “Whereas the activin receptor IIB decoy molecule used in the +/G610C and oim/oim studies neutralizes BMP9/10 in addition to activin and myostatin, a different activin receptor IIB decoy molecule that does not inhibit BMP9/10 was used in the Col1a1Jrt/+ mouse”: it is not clear what is the significance of this comparison.
  3. “These studies suggest that myostatin may serve as a novel therapeutic target in the treatment of OI although responses may be dose and mutation specific.”: it is perhaps difficult to say if it is mutation specific or not if the different mouse models have not been examined in a uniform manner.
  4. “The role of myostatin in muscle-bone crosstalk highlights this point. Inhibition of the myokine, myostatin, leads to altered mechanotransduction via increased muscle size in addition this inhibition is hypothesized to have a direct effect on biochemical signaling, as well, leading to a downregulation of osteoclast”: please improve sentence structure and content.

Response: We thank the reviewer for these comments that have led to a clearer description of myostatin inhibition in OI.

  1. Yes, both the oim and +/G610C mice were evaluated in the same study under the same conditions and protocol. Please see we have the description for clarification.

Lines 328-340: “Studies using activin receptor IIB decoy molecules suggest that the responsiveness to myostatin inhibition in OI maybe mutation specific. Specifically, in a large study with both +/G610C and oim/oim mice on the same congenic C57BL/6J background,  activin receptor IIB decoy molecule (RAP-031; Acceleron Pharma, Inc) treatment resulted in improved skeletal, but not muscle properties in the +/G610C mouse, modeling a mild to moderate human OI, and improved skeletal muscle contractile, but not skeletal properties in the oim/oim mouse (Jeong, Daghlas, Kahveci, et al.; Jeong, Daghlas, Xie, et al.). The activin receptor IIB decoy molecule (RAP-031) used in the +/G610C and oim/oim studies neutralizes activin and myostatin, but also thought to neutralize BMP9/10.  The ability to bind BMP9/10 was implicated in the vascular side effects seen in human trials using the human activin receptor IIB decoy molecule equivalent (ACE-031; Acceleron Pharma, Inc)(Attie et al.; Campbell et al.).  In a separate study, utilizing a different activin receptor IIB decoy molecule (ACE-2494; Acceleron Pharma, Inc) which does not neutralize BMP9/10, in the Col1a1Jrt/+ mouse led to increased muscle mass, bone mass, and improved bone geometry (Tauer and Rauch).  ”.

  1. Please see above where we have clarified the distinction between the activin receptor IIB decoy molecules (RAP-031, ACE-031, and ACE-2494).
  2. As indicated above, the studies using the activin receptor decoy molecule RAP-031 in the +/G610C and oim/oim mice were performed under the same conditions and the mice were on the same congenic background. The outcome measures differed between the two models, suggesting that these responses maybe mutation and/or severity specific.
  3. Lines 366-369: “The role of myostatin in muscle-bone crosstalk highlights this point. Inhibition of the myokine, myostatin, leads to increased muscle size and force, which is hypothesized to act through mechanotransduction to increase bone mass. Myostatin inhibition is also hypothesized to act directly through biochemical signaling, to a down regulate osteoclastogenesis (Qin et al.; Dankbar et al.).”

  1. Comment: Conclusions
  • Is there a recommendation that can be made about physical exercise in OI based on the review findings?

 Response: We agree that a more extensive conclusion regarding exercise in OI could be presented. Please see that we have added the following:

Lines 377-384: “Improving muscle strength and exploiting the muscle-bone crosstalk through exercise/WBV and novel drug therapies, respectively, promises to lead to improved patient outcomes and quality of life in OI. Although the studies regarding exercise in OI are limited, they suggest that children with type I OI tend to be just as active as their healthy peers, even though they exhibit reduced muscle strength (Pouliot-Laforte et al.). This, along with the recent study suggesting mice with OI require greater forces on bone to a elicit an osteogenic response (Berman et al.), further emphasize the need to explore different targeted physiotherapeutic modalities for the potential treatment of OI.” 

  1. Comment: General
  2. It would add to also discuss the production of collagen from osteoblasts as a result of physical activity which can affect bone quality.
  3. The paragraph “The role of these osteokines in osteogenesis imperfecta has been actively….”until “...that can exploit this relationship by improving muscle function to ultimately improve bone strength.” seems to have been written in a less structured, clear and concise manner compared to the rest of the article. In several places it is not clear what the significance of the stated literature is. Please revise accordingly.
  4. There is extensive description of the bone on the cellular level but this is absent for the muscles.
  5. The title is “Impact of intrinsic muscle weakness on muscle-bone crosstalk in osteogenesis imperfecta”: please elaborate more on the conclusions that can be made about the intrinsic properties of OI muscle; the article seems to focus a lot on the muscle-bone interaction, which is important in OI pathophysiology but not completely in line with the title. Also please include what is the role of collagen in muscle structure and function.

Response: We thank the reviewer for these general comments to help improve the overall quality of our manuscript. Please see that we have addressed each point below.

  1. There is a paucity of literature describing physical activity in relation to bone quality and collagen production by osteoblasts. The studies in rodent models is highly variable in outcome measures (for review, Portier, H.; Benaitreau, D.; Pallu, S. Does Physical Exercise Always Improve Bone Quality in Rats? Life 2020, 10, 217. https://doi.org/10.3390/life10100217), and we petition beyond the scope of our present review.
  2. Please see through the extensive rewrite from lines 319-349 that we have clarified the significance of this segment.
  3. Please see that we have expanded the discussion of the muscle molecular and cellular components in the comparisons of the OI muscle weakness to muscle molecular and cellular components in specific muscular dystrophies (Lines 168-186)
  4. Lines (375-384) Please see that we have expanded our conclusions as follows:

“Muscle-bone crosstalk is an important contributor to overall musculoskeletal health, and impaired muscle function in OI patients may further compromise their already compromised skeletal integrity. Muscle and bone communicate with one another via both mechanotransduction and biochemical signaling.  Improving muscle strength and exploiting the muscle-bone crosstalk through exercise/WBV and novel drug therapies, respectively, promises to lead to improved patient outcomes and quality of life in OI. Although the studies regarding exercise in OI are limited, they suggest that children with OI tend to be just as active as their healthy peers, even though they exhibit reduced muscle strength (Van Brussel et al.). This, along with the recent study suggesting mice with OI require greater forces on bone to a elicit an osteogenic response (Berman et al.), further emphasize the need to explore different targeted physiotherapeutic modalities for the potential treatment of OI.”

Round 2

Reviewer 2 Report

The manuscript has been significantly improved, nice added paragraph about collagen and muscle pathology in OI.